# Improvement of Gait after 4 Weeks of Wearable Focal Muscle Vibration Therapy for Individuals with Diabetic Peripheral Neuropathy

**DOI:** 10.3390/jcm9113767

**Published:** 2020-11-22

**Authors:** Josiah Rippetoe, Hongwu Wang, Shirley A. James, Carol Dionne, Bethany Block, Matthew Beckner

**Affiliations:** 1Department of Rehabilitation Sciences, College of Allied Health, University of Oklahoma Health Sciences Center, Oklahoma City, OK 73104, USA; Josiah-Rippetoe@ouhsc.edu (J.R.); shirley-james@ouhsc.edu (S.A.J.); Carol-Dionne@ouhsc.edu (C.D.); Bethany-Block@ouhsc.edu (B.B.); Matthew-Beckner@ouhsc.edu (M.B.); 2Peggy and Charles Stephenson School of Biomedical Engineering, University of Oklahoma, Norman, OK 73019, USA

**Keywords:** diabetic peripheral neuropathy, wearable focal muscles vibration, gait, spatiotemporal, kinematics, kinetics

## Abstract

People with diabetic peripheral neuropathy (DPN) experience lower quality of life caused by associated balance, posture, and gait impairments. While focal muscle vibration (FMV) has been associated with improvements in gait performance in individuals with neurological disorders, little is known about its effectiveness in patients with DPN. The purpose of this study was to investigate the effect of FMV on gait outcomes in patients with DPN. The authors randomized 23 participants into three FMV intervention groups depending upon the delivery of vibration. Participants applied wearable FMV to the bilateral quadriceps, gastrocnemius, and tibialis anterior, 10 min per muscle, three times per week over a four-week period. Spatiotemporal, kinematic, and kinetic gait parameters at baseline and post-intervention were calculated and analyzed. Gait speed, cadence, stride time, left and right stance time, duration of double limb support, and left and right knee flexor moments significantly improved after four weeks of FMV. Trends toward significant improvements were noted in maximum left and right knee flexion. Results indicate that FMV therapy was associated with improvements in gait parameters in individuals with DPN, warranting expanded study of FMV therapy for long-term gait performance improvement in these individuals.

## 1. Introduction

Up to 30% of individuals diagnosed with Type I or Type II Diabetes Mellitus will develop diabetic peripheral neuropathy (DPN) [1]. Accounting for approximately 27% of the US annual healthcare cost of diabetes, [2] which was $327 billion in 2017, [3] people with DPN experience a decline in their quality of life caused by the balance, posture, and gait impairments associated with DPN [4,5,6]. DPN causes metabolic and microvascular alterations in the peripheral nerves supplying the lower extremities [7,8]. These alterations create sensorimotor changes in the hands, fingers, feet, and toes, accompanied by debilitating pain [7,8]. The chronic nature of hyperglycemic exposure in people with diabetes, along with associated cardiovascular risk factors including obesity, hypertension, and cigarette smoking, all contribute to DPN [7]. One study reported that while their participants with DPN experienced a two-fold increase in healthcare costs, $12,492, those with the additional complications of DPN experienced a fourfold increase, up to $30,755 [9].

DPN-related tissue degradation of the sensorimotor system can negatively impact community participation. First, decreases in strength and motor unit activation in the muscles of the upper and lower extremities impair individuals’ ability to complete activities of daily living, and to walk functionally [10,11,12]. Because gait kinematics and kinetics become altered, reductions in speed, stride length, and cadence, along with increases in stance period and double limb support become apparent [5,13]. People with DPN also demonstrate reductions in knee and ankle joint range of motion while walking at self-selected and brisk speeds [5,11,14]. These impairments are linked to limited active ankle dorsiflexion during stance phase and swing phase, limited peak plantar flexion moment during terminal stance and push-off, and knee flexion achieved during pre-swing and swing phases of the gait cycle [5,11,14]. These gait deviations implicate muscle weakness of the gastroc-soleus complex from midstance through pre-swing, and the pre-tibial muscle group from initial swing through loading, permanently impacting ankle motion and ankle joint power (A2) [12,15,16,17,18].

Published state-of-the-art interventions for individuals with DPN include whole-body vibration (WBV) and physical exercise [19,20,21]. Researchers have reported improvements in balance, enhanced vibratory sensation, increases in muscle strength, and decreases in pain after WBV [22,23,24,25,26]. While Hong and colleagues [21] reported improvements in several spatiotemporal gait patterns in one individual treated with DPN, WBV has several associated side effects, including disruption of the skeletal, visual, and vestibular systems [27]. One study found that while physical exercise improves the symptoms of DPN, patients often find exercise regimens difficult to maintain over time [28].

Focal muscle vibration (FMV) applies targeted mechanical vibration to soft tissue, including muscles or muscle groups, mitigating potential negative effects associated with WBV, while providing the benefits of vibration therapy [29]. Researchers have published improvements in gait in individuals with stroke, Parkinson’s disease, and spinal cord injury [29]. FMV can stimulate the primary somatosensory cortex, facilitating its interneuron connections and enhancing motor control [29,30,31]. McKinney et al. assessed the impact of a wearable tactile feedback system for gait rehabilitation in patients with non-diabetic peripheral neuropathy. Even though the stimulus in the study was only used as a feedback system, most participants could sense the vibration [32]. While FMV provides benefits to both the peripheral system and neuromotor system, and increasing motor control improves gait [33], FMV has the potential to mitigate the gait deviations caused by DPN.

The purpose of this study was to investigate wearable FMV as an intervention to improve gait performance in individuals with DPN. We hypothesized that FMV would improve the spatiotemporal, kinematic, and kinetic gait parameters affected by DPN. We anticipate this study will provide preliminary evidence of this promising intervention in individuals with DPN.

## 2. Experimental Section

### 2.1. Participants

This pilot randomized parallel group clinical trial, conducted at the Technology for Occupational Performance (TOP) Laboratory and the Center for Human Performance Measurement (CHPM) at the University of Oklahoma Health Sciences Center (OUHSC), consisted of a convenience sample of 23 individuals recruited from the Oklahoma City Metropolitan area. Investigators screened consented participants between October 2018 and August 2020. Inclusion criteria included a medical diagnosis of diabetes of at least one-year duration, the presence of diabetic peripheral neuropathy as indicated by failing the ten-site 5.07 (10 g) monofilament sensory test [34], age of at least 18 years, ability to stand without the use of an assistive device, lack of other neurological or orthopedic diagnoses, normal vision (after correction as needed), and English language proficiency. Exclusion criteria included non-diabetic causes of neuropathy, symptomatic peripheral vascular disease, joint pain or swelling in the lower extremities that interferes with walking, and additional diagnoses that could interfere with walking, including amputation. The study protocol was approved by the University of Oklahoma Institutional Review Board (#9688).

### 2.2. Gait Measurement Methods

Investigators collected quantitative gait parameters using the Qualysis™ (Qualisys AB, Göteborg, Sweden) motion capture system and AMTI™ (Advanced Mechanical Technology, Inc., Watertown, MA, USA) force plates, placing reflective markers on 58 anatomical landmarks, using the Qualysis™ motion capture system’s 12 cameras (sampling rate 120 Hz) and the AMTI™ force plates (sampling rate 1 kHz) to capture gait performance data. Each participant first performed a static trial to develop their individual skeletal model, which the investigators then exported to Visual3D© (C-Motion, Germantown, MD, USA) for data analysis. Participants then walked at a self-selected speed for 25 feet for three to five trials, using the normal pace they used for everyday life. Investigators then calculated kinematic and kinetic data using Visual3D©.

### 2.3. Gait Outcome Measures

Investigators analyzed the following gait parameters: gait speed, stride length, stride width, stride time, left and right stance time, left and right swing time, cadence, duration of double support, left and right peak knee flexion, left and right peak dorsiflexion, left and right peak plantarflexion, left and right peak knee flexor moment, left and right peak plantar flexor moment, and left and right peak ankle joint power. Visual3D© [35] defines the spatiotemporal measures as follows: gait speed is the stride length divided by stride time, stride length is the distance from heel strike to heel strike on the same side, stride time is the duration of the gait cycle from heel strike to heel strike on the same side, left stance time is the time from left heel strike to left toe off, right stance time is the time from right heel strike to right toe off, left swing time is the time from left toe off to left heel strike, right swing time is the time from right toe off to right heel strike, mean cadence is the average number of steps taken per minute, and duration of double support is the amount of time spent with both feet in contact with the surface.

### 2.4. Intervention Protocol

Investigators collected demographic (age, weight, height, duration in years of type II diabetes mellitus, gender, and ethnicity) and baseline gait data at the first visit. FMV intervention included a four-week, at-home intervention using a modified version of a commercially available wearable FMV (Myovolt™, Christchurch, New Zealand) device, one for each leg. The modified version consisted of one vibration motor with the recommended amplitude of 1.2 mm. Investigators asked participants to wear the device over their typical clothes, and use the device as they preferred during their daily activities. The Myovolt™ devices use three different 120 Hz frequency modes; sinusoidal (Mode 1), pulsing (Mode 2), and continuous (Mode 3). Investigators assigned participants to one of the three mode intervention groups using permutated blocks of three or six (Figure 1).

Investigators selected three vibration modes to determine whether the method of vibration delivery would impact changes in gait. Investigators instructed the participants to apply FMV devices to the tibialis anterior, quadriceps, and gastrocnemius muscles of each leg (Figure 2) for 10 min each, three times per week over a four-week period. Participants completed an intervention log. Following the intervention, investigators re-evaluated gait patterns using a protocol identical to the baseline protocol.

### 2.5. Data Analysis

Investigators analyzed data using Visual 3D© and SPSS Statistics 25 (IBM, Armonk, NY, USA). First, investigators calculated summary statistics for each gait parameter (spatiotemporal parameters, joint angles, and joint moments) pre and post-intervention. Investigators analyzed normality of data using the Kolmogorov-Smirnov test, and used paired t-tests when appropriate to evaluate within group changes. For non-normally distributed data, investigators used Wilcoxon signed-rank tests. Investigators then used two-way mixed analysis of variance (ANOVA) to examine how the changes in gait parameters differed between different vibration groups, and differed between individuals who use mobility related assistive technology (AT) and those who do not use AT devices. Investigators analyzed all data using SPSS Statistics 25 with an alpha = 0.05.

## 3. Results

While investigators screened a total of 24 subjects, one individual declined to participate, making our total enrollment 23 (Figure 1). Seven participants utilized either a cane or a walker during the data collection, which affected their kinetic data and so were excluded from the kinetic data analysis (Figure 1). Each participants’ demographic information is contained in Table 1. No statistical difference in age, weight, height, and number of years with diabetes were found among the three vibration groups (Table 1).

Six spatiotemporal parameters demonstrated significant improvements from pre-test to post-test. Gait speed increased by 0.09 (0.03, 0.18) seconds, stride time decreased by 0.08 (95% CI = −0.14, −0.04) seconds, cadence increased by 6.5 (95% CI = 3.6, 10.2) steps per minute, left stance time decreased by 0.06 (95% CI = −0.11, −0.01) seconds, right stance time decreased by 0.08 (95% CI = −0.13, −0.02) seconds, and double limb support decreased by 0.05 (95% CI = −0.09, 0.00) seconds (Table 2). Spatiotemporal gait parameters did not significantly change based on the three modes of vibration. Changes in the spatiotemporal gait parameters were larger in Mode 1 and 2 than Mode 3.

Trends toward significant changes were found in left (*p* = 0.083) and right (*p* = 0.078) peak knee flexion. There were significant increases in left and right peak knee flexor moment (Table 3). Between-group analysis indicated that mode of vibration was not significantly associated with changes in kinematics and kinetics but similarly, changes in Mode 1 and Mode 2 were higher than Mode 3.

Of the twenty-three participants, seven used mobility related assistive devices. The non-AT group had significantly faster gait speeds (*p* = 0.028), shorter stride time (*p* < 0.001), higher cadence (*p* < 0.001), shorter left (*p* = 0.001) and right (*p* < 0.001) stance time, shorter left (*p* < 0.001) and right (*p* = 0.019) swing time, and shorter double limb support (*p* = 0.001). Right peak knee flexion was significantly different between the AT and the non-AT group (*p* = 0.036) as shown in Table 4. When compared, although changes in gait parameters between individuals using AT and non-AT were not significant, AT users showed higher improvements in gait speed, stride width, stride time, left and right stance time, left swing time, and double limb support. The AT group showed more improvement in left and right peak knee flexion, while more changes were observed in the non-AT group on left and right dorsiflexion, as well as left and right plantarflexion.

Participants logged the number of sessions or days they used the FMV device in a chart prepared by the investigators. We calculated participant compliance as the ratio of the real to the recommended number of sessions. Two out of the 23 participants did not provide their device use log. The compliance for the remaining 21 participants was 100%. No participant was involved in other interventions or exercise therapy for the duration of this study, but were asked to follow their typical daily routines. One participant noted there were no interventions available to help him with his DPN. There is currently no standard care for DPN. Eight participants commented they walked more or were more active after two weeks of vibration therapy.

## 4. Discussion

Other studies have reported that FMV therapy improves gait in individuals with diagnoses of stroke, Parkinson’s disease, and spinal cord injury [29], however no studies have investigated the effects of FMV on gait in individuals with DPN. In this study, investigators utilized FMV therapy for specific lower extremity muscles that impacted gait performance for 30 min, three times per week for four consecutive weeks. This is the first study to investigate how intervention with FMV impacts gait in individuals with DPN. The gait parameters of this study at baseline were consistent with other studies, specifically of gait speed, stride length, stride time, maximum knee flexion, maximum dorsiflexion, maximum plantarflexion, and peak plantar flexor moment [11,12,36,37]. The participants showed excellent compliance with the wearable FMV therapy, which indicated that the intervention was well tolerated. Because participants logged their own device use, their reports may be subject to recall bias. The improved technology we are developing will enable us to track device use with a cellular telephone application. In our next study, we also plan to track the physical mobility of the participants via a wearable device such as Fitbit to see how their physical activity changes during the study intervention period.

The results of this study demonstrated that a four-week intervention of FMV can enhance the spatiotemporal parameters of gait, similar to studies reported for individuals with Parkinson’s disease (PD) and multiple sclerosis (MS) [38,39]. Camerota et al. [38] found that after three consecutive days of one 60-min session of FMV per day, patients with PD had increased gait speed, stride length, and step length. Camerota et al. [39] also showed that after three consecutive days of one 60-min session of FMV per day patients with MS had significant improvement in gait speed, double limb support, step length, stride length, and cadence. The results were also similar to Melai et al. [40] who utilized a 24-week exercise program to strengthen the muscles of the lower extremities of individuals with DPN. The exercise program improved several spatiotemporal gait parameters including stride length and gait speed [40].

After four weeks of FMV therapy, the decrease in duration of double support may have reduced the stance time, thereby, decreasing the stride time while the swing time did not change. Consequently, both cadence and gait speed were then positively impacted even though the stride length did not significantly improve. The improvement in the spatiotemporal gait parameters may have been because the FMV improved proprioception, a specific target to enhance gait [41]. Peppe et al. [42] found that FMV improved proprioception in individuals with Parkinson’s disease. However, the joint kinematics and kinetics may not have improved due to the gastric-soleus complex not receiving adequate vibration therapy. Filippi et al. [43] and Pazzaglia et al. [31] utilized a protocol that consisted of FMV therapy applied to the quadriceps and the triceps surae muscles for three consecutive days to cumulate the after-effects of FMV therapy. Feltroni et al. [44] utilized a 30-min therapy to the quadriceps muscle over five consecutive days, which strengthened the quadriceps muscle. Furthermore, the authors of all three studies instructed participants to isometrically contract the muscles during the FMV therapy to provide greater proprioceptive input and enhance the muscle activation [31,43,44]. Therefore, the protocol of this study, which consisted of three days per week FMV therapy but not necessarily consecutive days, only stimulating each muscle for 10 min each session, and not instructing participants to contract the muscles being stimulated during the FMV session, may not have produced the cumulative effects to enhance the strength of the quadriceps, tibialis anterior, or gastrocnemius muscles, thereby, not improving the joint kinematics and kinetics.

Peak knee flexor moments significantly improved and maximum knee flexion angles of both lower extremities were trending toward significance increases. According to a systematic review comparing gait kinetics and kinematics among diabetics with neuropathy, diabetics without neuropathy and non-diabetes individuals, maximum knee flexion angle was significantly higher in the non-diabetes group [12]. Both left and right maximum knee flexion angles increased by about 4 degrees, which indicated a more forceful push off. In individuals with DPN, the smaller knee flexion reinforces the hypothesis of a poor eccentric control because they assume a posture that saves quadriceps effort [45]. The changes on the left (mean from pre to post: −2.5 to −0.24 degree) and right (mean from per to post: −1.85 to 1.37 degree) maximum knee extension angles were not statistically significant, but it is evidence that the knee hyperextension was decreased, which is a result of gait deviations and muscle weakness in patients with DPN [41]. Maximum dorsiflexion and plantarflexion stayed pretty consistent before and after the intervention, this may be partially due to the ankle not having much movement to begin with. The improvements in the ankle power, though not significant, indicated increased strength and power generated during push off from the plantar flexors, which might explain the overall better gait and improvements in the spatiotemporal parameters.

The baseline gait parameters of this study are comparable to those of other studies investigating gait in patients with DPN. In a study comparing walking stability and sensorimotor function in older adults with DPN and age-matched controls, the average gait speed for the DPN groups were 0.84 and 0.98 m/s with a cadence of 90.3 and 99.2 [46]. The study by Suzuki et al. 2019 on the effect of exercise with rhythmic auditory stimulation on gait in patients with DPN had higher baseline gait speed (1.29 m/s), and cadence (124.25) than this study, along with smaller step length (0.62 m). This could be due to the younger mean age (59 years) of their participants, and their mean shorter height (63.76′′) [47]. Saleh and Rehab 2014 assessed ankle proprioceptive training for patients with DPN; their participants had slower baseline speed (0.71 m/s), lower cadence (80), similar stride time (1.24 s), smaller step length (0.52 m), and smaller double support time (0.32 s) than our baseline data [48]. Another study examining the efficacy of an exercise rehabilitation program in improving the gait of patients with DPN had slower baseline speed (0.71 m/s), smaller cadence (82.94), and larger stride time (1.72 s), but with similar double limb support time (0.46 s) and ankle range of motion (19.38 degree) when compared to this study [49]. We did not find significant outcome measure changes between the three intervention groups using the three different vibration modes. After intervention, we observed the changes in the gait parameters were larger in group 1 with the sinusoidal vibration and 2 with the pulsing vibration than group 3 with continuous vibration as shown in Table 2 and Table 3. This could be because of the better overall baseline values in most gait parameters in group 3. It could also be possible that DPN patients responded differently to the three type of vibration modes. Here we hypothesize the latter, as patients with DPN experience sensory loss, particularly vibratory sensation. When the continuous vibration was delivered in group 3, during initial use, it might work well but when the participants get used to the vibration, the effect was reduced or diminished. Alternatively, groups 1 and 2 with the changing vibratory frequencies may have taken longer for accommodation to the vibration pattern. Future study will be needed to test the hypothesis and if true, then the best vibration delivery could be randomized frequencies within a safe and effective range. However, group 3 (*n* = 10) had significantly higher baseline values on several spatiotemporal parameters (speed, stride length, left and right step length, left and right duration of double support) than group 1 (*n* = 7) and 2 (*n* = 6). With the relatively small sample size for each group, no conclusions can be drawn regarding the impact of mode of vibration on gait. Furthermore, we did not collect data on severity of peripheral neuropathy. This could contribute to the non-significant changes across the three groups. We will assess the severity of peripheral neuropathy (for example, the Michigan Neuropathy Screening Instrument) in our future studies at both baseline and post-intervention, as well as electrophysiology measures for sensation.

While both AT and non-AT users showed significant improvements, we observed greater changes in most of the spatiotemporal parameters in AT users than non-AT users. This could because that at the baseline, the AT users were walking slower with more compromised parameters. Given the wearable FMV does not need much active involvement with the users during the treatment, it could be more a viable option for those DPN patients with limited function so they can participate in other exercise interventions. The AT users group showed a trend to improve dorsiflexion compared to the non-AT group. The improvement of knee flexion was driven by the changes in knee flexion. This highlights the differences in gait between different types of patients. In a future study, we will improve the current method in gait data collection to be able to analyze gait kinetics for AT users, which will allow us to further learn how FMV works for AT users.

This study had several limitations. Investigators cannot exclude a placebo effect, since no control group was included in this study. We reviewed articles with control groups on other interventions for gait rehab, for example, exercise, to see how gait of the control group in other studies changed. In studies by Suzuki et al., 2019, both intervention and control groups demonstrated significant improvements in spatiotemporal parameters. However, the control group in their study underwent two weeks of supervised rehabilitative treatment (40 min/day), while the intervention group received extra rhythmic auditory stimulation [47]. Similarly, Saleh and Rehab reported significant improvement in gait parameters for both the intervention and control groups. However, their control group received traditional physical therapy exercise [48]. In a study where the control group received no intervention but regular activities, no significant improvement in any gait parameters were reported and some of the parameters got worse [49]. The authors did not include a control group because it is difficult to blind participants, and because the participants did not want to be in the control group with no vibration, especially after they experienced FMV during their first visits. The authors had improved the FMV technology to be able to deliver no effect sham vibrations (very low frequency with sound effect). Future study with better methodology, such as cross-over design will enable comparisons on effects of FMV between intervention and control groups. This study had a small sample size, which could lead to type II error, especially given the amount of the gait parameters we were examining. The authors are planning an expanded study to include a larger number of participants. In this pilot study, patients with a history of foot ulcers were not excluded, although no participants had active foot ulcers. We did not collect clinical parameters of diabetes severity such as HbA1C. These clinical parameters could affect the effect of FMV on gait. In future studies, we will assess the medical record of participants, and track their HbA1C at both baseline and post-intervention. Other joint angles, moments, and power may not have significantly improved after the four-week intervention due to two participants failing to comply with the FMV protocol and another participant experiencing an injury outside the study that significantly affected their gait. Investigators also allowed the participants to use their assistive device during the gait evaluation, which limited the analysis of the kinetic gait data. Future use of an over-head track harness system during gait trials will address this limitation. The FMV protocol in this study may not have been adequate to improve joint kinematics and kinetics. Future studies should utilize a protocol that could account for the cumulative effects of FMV therapy and allow for greater muscular control during the FMV therapy, thereby improving lower extremity muscular strength and joint kinematics and kinetics.

## 5. Conclusions

This pilot study demonstrated emerging evidence that FMV therapy is associated with improvements in some of the gait parameters in individuals with DPN. Results are preliminary, and indicate the feasibility of further study of this association with a larger sample size and with a control group.

## Figures and Tables

**Figure 1 jcm-09-03767-f001:**
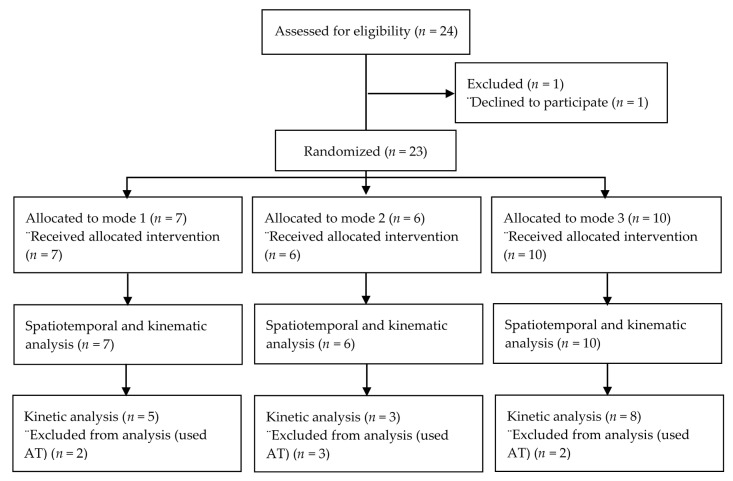
CONSORT Diagram. AT: Assistive Technology.

**Figure 2 jcm-09-03767-f002:**
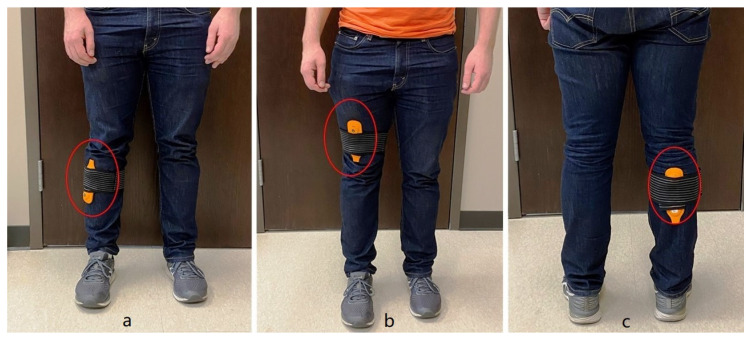
Attachment of MyoVolt™ focal muscle vibration device. (**a**) device attached using a strap to the right anterior tibialis muscle. (**b**) device attached using a strap to the right quadricep muscle. (**c**) device attached using a strap to the right gastrocnemius muscle.

**Table 1 jcm-09-03767-t001:** Patient Demographics: Age, weight, height, and length of diabetes diagnosis are all listed as mean (standard deviation) while number, sex, and race are listed as N.

Demographic Variables	All	Group 1	Group 2	Group 3	*p*-Value ^a^
Number	23	7	6	10	-
Age (years)	66.74 (10.76)	66.71 (13.35)	66.50 (5.75)	66.90 (12.08)	0.53
Weight (lbs.)	219 (58.14)	194.14 (44.21)	240.33 (62.49)	223.60 (63.11)	0.36
Height (inches)	67.46 (3.58)	65.86 (3.90)	67.50 (4.28)	68.55 (2.77)	0.39
Length of diabetes diagnosis (years)	17.78 (8.45)	16.43 (5.56)	21 (13.31)	16.90 (6.69)	0.58
Sex (F/M)	14/9	5/2	3/3	6/4	-
**Race**					
Caucasian	21	6	6	9	-
African-American	1	0	0	1	-
Euro-Asian	1	1	0	0	-

^a^ ANCOVA results.

**Table 2 jcm-09-03767-t002:** Mean (std) in spatiotemporal parameters for all participants and for the three vibration mode groups.

Parameter m = meters s = second		All Participants (*n* = 23)	Participants Receiving Mode 1 (*Sinusoidal Vibration*) (*n* = 7)	Participants Receiving Mode 2 (*Pulsing Vibration*) (*n* = 6)	Participants Receiving Mode 3 (*Continuous Vibration*) (*n* = 10)
Gait speed (m/s)	pre	0.82 (0.38)	0.68 (0.23)	0.58 (0.26)	1.07 (0.40)
	post	0.90 (0.31) ^ac^	0.85 (0.30)	1.02 (0.30)	0.90 (0.31)
Stride length (m)	pre	0.93 (0.32)	0.77 (0.38)	0.80 (0.18)	1.13 (0.23)
	post	1.03 (0.26) ^b^	1.04 (0.22)	0.93 (0.19)	1.09 (0.31)
Stride width (m)	pre	0.18 ± 0.04	0.18 (0.04)	0.20 (0.04)	0.18 (0.03)
	post	0.19 ± 0.03 ^b^	0.18 (0.03)	0.19 (0.04)	0.19 (0.03)
Stride time (s)	pre	1.32 (0.31)	1.37 (0.33)	1.51 (0.42)	1.16 (0.10)
	post	1.22 (0.26) ^ac^	1.30 (0.32)	1.35 (0.33)	1.08 (0.04)
Cadence (steps/min)	pre	95.04 (17.40)	91.22 (19.36)	84.97 (21.39)	103.76 (8.35)
	post	102.0 (16.3) ^ac^	96.73 (18.55)	94.49 (22.09)	110.08 (4.21)
Left stance time (s)	pre	0.88 (0.29)	0.94 (0.26)	1.09 (0.37)	0.71 (0.11)
	post	0.82 (0.21) ^ac^	0.86 (0.24)	0.94 (0.29)	0.71 (0.05)
Right stance time (s)	pre	0.86 (0.31)	0.87 (0.34)	1.06 (0.42)	0.74 (0.08)
	post	0.80 (0.21) ^bc^	0.85 (0.24)	0.88 (0.29)	0.70 (0.03)
Left swing time (s)	pre	0.43 (0.08)	0.43 (0.08)	0.42 (0.05)	0.43 (0.09)
	post	0.40 (0.06) ^a^	0.43 (0.07)	0.41 (0.06)	0.37 (0.04)
Right swing time (s)	pre	0.43 (0.08)	0.45 (0.12)	0.44 (0.06)	0.42 (0.05)
	post	0.42 (0.06) ^b^	0.44 (0.06)	0.44 (0.08)	0.39 (0.04)
Double limb support (s)	pre	0.47 (0.25)	0.51 (0.21)	0.66 (0.36)	0.32 (0.06)
	post	0.42 (0.20) ^ac^	0.43 (0.19)	0.54 (0.29)	0.33 (0.06)

^a^ Wilcoxon, ^b^ paired *t* test, ^c^
*p* < 0.05.

**Table 3 jcm-09-03767-t003:** Mean (std) in kinematic and kinetic parameters for all participants and for the three vibration mode groups.

Parameter L = Left; R = Right		Total (*n* = 23)	Mode 1 (*Sinusoidal Vibration*) (*n* = 7)	Mode 2 (*Pulsing Vibration*) (*n* = 6)	Mode 3 (*Continuous Vibration*) (*n* = 10)
L peak knee flexion (°)	pre	53.16 (9.31)	53.13 (9.79)	49.29 (7.52)	55.50 (10.02)
	post	57.10 (9.31) ^a^	56.67 (11.99)	56.15 (9.02)	57.96 (8.31)
L peak dorsiflexion (°)	pre	19.70 (4.34)	22.83 (1.91)	17.80 (7.25)	18.64 (1.79)
	post	19.32 (6.03) ^b^	21.15 (1.13)	16.25 (10.71)	19.89 (3.88)
L peak plantarflexion (°)	pre	−12.37 (8.68)	−9.02 (7.51)	−15.84 (8.84)	−12.64 (9.30)
	post	−12.70 (9.26) ^a^	−13.81 (6.84)	−12.84 (11.63)	−11.84 (10.08)
R peak knee flexion (°)	pre	52.09 (9.51)	53.68 (8.65)	48.16 (8.85)	53.34 (10.66)
	post	56.22 (11.92) ^b^	58.99 (11.94)	48.59 (16.14)	58.85 (11.92)
R peak dorsiflexion (°)	pre	21.12 (2.87)	21.36 (0.83)	23.71 (3.09)	19.40 (2.57)
	post	19.49 (6.32) ^b^	19.76 (2.33)	16.89 (11.53)	20.86 (3.66)
R peak plantarflexion (°)	pre	−11.50 (10.62)	−12.75 (9.53)	−9.21 (15.21)	−12.00 (9.04)
	post	−12.87 (8.15) ^a^	−12.99 (4.54)	−17.38 (9.70)	−10.09 (8.15)
		**Total (*n* = 16)**	**Mode 1 (*sinusoidal vibration*) (*n* = 5)**	**Mode 2 (*pulsing vibration*) (*n* = 3)**	**Mode 3 (*continuous vibration*) (*n* = 8)**
L peak knee flexor moment (N∙m/Kg)	pre	−0.61 (0.35)	−0.68 (0.39)	−0.61 (0.17)	−0.56 (0.40)
	post	−0.78 (0.35) ^a,c^	−0.94 (0.24)	−0.76 (0.06)	−0.68 (0.44)
L peak plantar flexor moment (N∙m/Kg)	pre	1.08 (0.18)	1.16 (0.15)	0.87 (0.07)	1.11 (0.18)
	post	1.17 (0.31) ^b^	1.08 (0.22)	1.29 (0.52)	1.17 (0.30)
L peak ankle power (W/Kg)	pre	2.24 (0.90)	2.48 (1.09)	1.70 (0.68)	2.29 (0.88)
	post	2.54 (1.02) ^b^	2.88 (0.95)	1.91 (1.19)	2.56 (1.03)
R peak knee flexor moment (N∙m/Kg)	pre	−0.57 (0.34)	−0.74 (0.23)	−0.67 (0.21)	−0.43 (0.40)
	post	−0.66 (0.41) ^a,c^	−0.81 (0.20)	−0.81 (0.10)	−0.51 (0.53)
R peak plantar flexor moment (N∙m/Kg)	pre	1.08 (0.23)	0.97 (0.22)	0.94 (0.02)	1.19 (0.22)
	post	1.08 (0.16) ^b^	1.07 (0.14)	0.98 (0.06)	1.13 (0.19)
R peak ankle power (W/Kg)	pre	2.19 (0.97)	1.91 (0.82)	1.61 (0.79)	2.58 (1.03)
	post	2.45 (0.92) ^b^	2.50 (0.57)	1.96 (0.93)	2.61 (1.12)

^a^ Wilcoxon, ^b^ paired t test, ^c^
*p* < 0.05.

**Table 4 jcm-09-03767-t004:** Mean (std) in spatiotemporal and kinematic parameters for the non-AT and AT users.

Parameter m = meters s = second		Non-AT User (*n* = 16)	AT User (*n* = 9)
Gait speed (m/s)	pre	0.93 (0.35)	0.58 (0.35)
	post	0.98 (0.25)	0.71 (0.37)
Stride length (m)	pre	0.97 (0.34)	0.86 (0.28)
	post	1.07 (0.26)	0.95 (0.27)
Stride width (m)	pre	0.18 (0.04)	0.19 (0.03)
	post	0.19 (0.03)	0.19 (0.04)
Stride time (s)	pre	1.17 (0.10)	1.65 (0.37)
	post	1.10 (0.08)	1.48 (0.34)
Cadence (steps/min)	pre	103.00 (8.60)	76.84 (19.10)
	post	108.80 (7.52)	86.29 (20.54)
Left stance time (s)	pre	0.77 (0.09)	1.13 (0.42)
	post	0.72 (0.07)	1.03 (0.28)
Right stance time (s)	pre	0.73 (0.06)	1.16 (0.39)
	post	0.70 (0.06)	1.01 (0.27)
Left swing time (s)	pre	0.40 (0.04)	0.50 (0.09)
	post	0.38 (0.03)	0.44 (0.08)
Right swing time (s)	pre	0.41 (0.05)	0.48 (0.11)
	post	0.40 (0.,04)	0.46 (0.09)
Double limb support (s)	pre	0.37 (0.07)	0.69 (0.37)
	post	0.33 (0.06)	0.61 (0.25)
L peal knee flexion (°)	pre	54.18 (8.94)	50.83 (10.41)
	post	59.60 (6.86)	51.38 (12.05)
L peak dorsiflexion (°)	pre	20.45 (2.56)	17.98 (6.91)
	post	20.69 (3.59)	16.19 (9.20)
L peak plantarflexion (°)	pre	−14.58 (6.49)	−7.31 (11.32)
	post	−13.81 (8.76)	−10.17 (10.58)
R peak knee flexion (°)	pre	53.87 (9.61)	48.03 (8.53)
	post	59.74 (7.32)	48.15 (16.67)
R peak dorsiflexion (°)	pre	20.86 (2.10)	21.74 (4.31)
	post	21.04 (3.05)	15.95 (10.10)
R peak plantarflexion (°)	pre	−13.57 (9.20)	−6.77 (12.83)
	post	−11.56 (7.61)	−15.87 (9.13)
L peak knee flexor moment (N·m/Kg)	pre	−0.61 (0.35)	*
	post	−0.78 (0.35)	*
L peak plantar flexor moment (N·m/Kg)	pre	1.08 (0.18)	*
	post	1.17 (0.31)	*
L peak ankle power (W/Kg)	pre	2.24 (0.90)	*
	post	2.54 (1.02)	*
R peak knee flexor moment (N∙m/Kg)	pre	−0.57 (0.34)	*
	post	−0.66 (0.41)	*
R peak plantar flexor moment (N∙m/Kg)	pre	1.08 (0.23)	*
	post	1.08 (0.16)	*
R peak ankle power (W/Kg)	pre	2.19 (0.97)	*
	post	2.45 (0.92)	*

* kinetic data for AT users were not available due to interference between the AT devices and the markers for the 3D motion system.

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
