# Peer review of "Improvement of Gait after 4 Weeks of Wearable Focal Muscle Vibration Therapy for Individuals with Diabetic Peripheral Neuropathy"

_jcm, 2020, doi:10.3390/jcm9113767_

Round 1

Reviewer 1 Report

The submitted manuscript reports the initial findings from a trial of muscle vibration therapy in diabetic peripheral neuropathy (DPN). This therapeutic approach could be valuable in improving gait and physical function in patients with DPN. The advantages of the treatment are that this is a home-based intervention and that it can be applied to multiple areas of the lower extremity. This is a strength of the manuscript that could be emphasized further. The study outcomes show significant improvements in gait spatiotemporal parameters of stride time reduction and increased cadence and kinetic parameter of knee flexor moments.  The weaknesses include a lack of control group and limited description of the intervention and compliance.

Areas of clarification include the following:

  1. How was compliance to the training verified (e.g. video, chat, etc) and what was the compliance? A log is introduced but the results of the log are unknown. Also, were subjects involved in any other intervention or exercise therapy or normal walking.
  2. How were participants instructed to wear the vibration trainer (on skin, over bulky clothes like jeans, etc). This seems like an important variable that would affect the overall vibration therapy received.
  3. What was the amplitude of the vibration and what type of MyoVolt was used? The products listed from this company do not align with the product shown in Figure 2 unless it is the wrist/elbow device.
  4. Were the subjects instructed to stand during treatment and was this recorded in the log? if there were standing, were they wearing shoes? As this approach is somewhat novel it may be helpful to standardize these elements moving forward especially as they relate to somatosensory changes.
  5. Including several subjects with assistive devices confounds the study. Given that there were no differences between the intervention groups, it is suggested to split the study into 2 groups or report on a subset of outcome measures. Was this therapy more or less helpful for those with assistive devices?
  6. The testing conditions could use further clarification. What were the instructions given during the gait assessments? Were the subjects told to walk at normal pace or to walk as fast as they were comfortable? A self-selected gait is open for interpretation as one might choose a slower or faster gait especially as the study was designed to investigate improvements.
  7. There is a bit of back and forth with the data reporting/discussion in that an ANCOVA is introduced, but no individual groups results are shown. Then, in the discussion page 7, line 218-224, group differences are referenced. If this is to remain a focus, at least some of these individual group results should be reported.
  8. An attempt should be made to compare baseline gait with other studies of DPN.
  9. Are there any other clinical parameters to include including history of foot ulcer and HbA1C?
  10. Ankle range of motion did not improve with intervention. Is this because the ROM was similar to a control sample? The lack of improvement is discounted in the discussion and it is unclear if this is intended to mean that the ankle ROM improvement is hard to detect. It is likely that is was not compromised to begin with. This ties in with (9) above.

There are several slight grammar issues throughout that should be revisited.

Author Response

1.How was compliance to the training verified (e.g. video, chat, etc) and what was the compliance? A log is introduced but the results of the log are unknown. Also, were subjects involved in any other intervention or exercise therapy or normal walking.

Thank you for this comment.

Those statements were added in the result session:

“Participants logged the number of sessions or days they used the FMV device in a chart prepared by the investigators. We calculated participant compliance as the ratio of the real to the recommended number of sessions. Two out of the 23 participants did not provide their device use log. The compliance for the remaining 21 participants was 100%. No participant was involved in other interventions or exercise therapy for the duration of this study, but were asked to follow their typical daily routines. One participant noted there were no interventions available to help him with his DPN. There is currently no standard care for DPN. Several of the participants commented they walked more or were more active after two weeks of vibration therapy.”

Those statements were added to the discussion session:

“The participants showed excellent compliance with the wearable FMV therapy which indicated the intervention was well tolerated. Because participants logged their own device use, their reports may be subject to recall bias. The improved technology we are developing, will enable us to track device use with a cellular telephone application. In our next study, we also plan to track the physical mobility of the participants via a wearable device such as Fitbit to see how their physical activity changes during the study intervention period.”

2.How were participants instructed to wear the vibration trainer (on skin, over bulky clothes like jeans, etc). This seems like an important variable that would affect the overall vibration therapy received.

We appreciate your comments and agree this could be a confounding factor as the vibration delivered to the target muscles would be affected by the clothes. After reviewed the log from the participants, the only comment made was that one subject felt that the straps for the wearable FMV worked better when the device was used over clothes than when they used the device against just their skin. In this study the participants were instructed to wear the device over their typical clothes. Below statements were added in the methodology:

“Investigators asked participants to wear the device over their typical clothes.”

Those statements were added to the discussion session:

“In this study the participants were instructed to wear the device over their typical clothes. The authors admitted that the vibration delivered to the target muscles would be affected by the type of clothes. Future study will be needed to investigate whether vibration over clothes impacts the acute effect of the vibration, and log how the device is worn by the participants.”

3.What was the amplitude of the vibration and what type of MyoVolt was used? The products listed from this company do not align with the product shown in Figure 2 unless it is the wrist/elbow device.

Thank you for the comment. We used the modified version (modified by the original manufacture) of the commercially available Myovolt device with only one motor with the amplitude recommended: 1.2mm. Those changes were made to the methodology session:

“FMV intervention included a four-week, at-home intervention using a modified version of a commercially available wearable FMV (Myovolt™, Christchurch, New Zealand) device, one for each leg. The modified version consisted of one vibration motor with the recommended amplitude of 1.2 mm.”

4.Were the subjects instructed to stand during treatment and was this recorded in the log? if there were standing, were they wearing shoes? As this approach is somewhat novel it may be helpful to standardize these elements moving forward especially as they relate to somatosensory changes.

Thank you for the comment. As a wearable device used during activities of daily living, we instructed the subjects to use the device as they were able to during their activities. While this will impact how our results compare to the literature, we feel this was actually a strength as in most of previous FMV studies the participants were in a supine position. Some participants did log how they used the device, with majority of them applying the vibration while sitting or lying down. No participants indicated they were doing anything active while wearing the device. We did not track whether they were wearing the shoes during the application of the vibration. Most common responses included they were watching television or reading while they were using the device. We agree with the reviewer that a standardized protocol of these elements could be very beneficial but in this pilot study we were not able to investigate those factors, and in the literature, there is lack of information on how those factors would impact the effect of the vibration therapy.

5.Including several subjects with assistive devices confounds the study. Given that there were no differences between the intervention groups, it is suggested to split the study into 2 groups or report on a subset of outcome measures. Was this therapy more or less helpful for those with assistive devices?

Thank you for the suggestion. We have added Table 4:  the results for the AT group versus the non-AT group. These statements were added in the result session:

“Of the twenty-three participants, seven used mobility related assistive devices. The non-AT group had significantly faster gait speeds (p=0.028), shorter stride time (p<0.001), higher cadence (p<0.001), shorter left (p=0.001) and right (p<0.001) stance time, shorter left (p<0.001) and right (p=0.019) swing time, and shorter double limb support (p=.001). Right peak knee flexion was significantly different between the AT and the non-AT group (p=0.036). When compared, although changes in gait parameters between individuals using AT and non-AT were not significant, AT users showed higher improvements in gait speed, stride width, stride time, left and right stance time. left swing time, and double limb support. The AT group showed more improvement in left and right peak knee flexion, while more changes were observed in the non-AT group on left and right dorsiflexion, as well as left and right plantarflexion.”

6.The testing conditions could use further clarification. What were the instructions given during the gait assessments? Were the subjects told to walk at normal pace or to walk as fast as they were comfortable? A self-selected gait is open for interpretation as one might choose a slower or faster gait especially as the study was designed to investigate improvements.

Thank you for the comment. This statement was added to the testing condition session:

“Participants then walked at a self-selected speed for 25 feet for three to five trials, using the normal pace they used for everyday life.”

7.There is a bit of back and forth with the data reporting/discussion in that an ANCOVA is introduced, but no individual groups results are shown. Then, in the discussion page 7, line 218-224, group differences are referenced. If this is to remain a focus, at least some of these individual group results should be reported.

Thank you for the comment. We have now included pre- and post- mean and SD of the gait parameters for each group as shown in the revised Table 2 and 3. Actually, we should run a 2 by 3 mixed ANOVA instead of ANCOVA. These changes were made in methodology session:

“Investigators then used two-way mixed analysis of variance (ANOVA) to examine how the changes in gait parameters differed between different vibration groups, and differed between individuals who use mobility related assistive technology (AT) and those who do not use AT devices.”

8.An attempt should be made to compare baseline gait with other studies of DPN.

Thank you for the suggestion. We have strengthened this portion of the paper. These statements were added in the discussion session:

“The baseline gait parameters of this study are comparable to those other studies investigating gait of DPN. In a study comparing walking stability and sensorimotor function in older adults with DPN and age-matched controls, the average gait speed for the DPN groups were 0.84 and 0.98 m/s with cadence of 90.3 and 99.2 [47]. The study by Suzuki et al. 2019 on effect of exercise with rhythmic auditory stimulation on gait in patients with DPN had higher baseline gait speed (1.29 m/s), cadence (124.25) than this study with smaller step length (0.62 m). This could be due to the younger ages (59 years)) of their participants, and shorter height (63.76”) [48]. NI et al., 2014 assessed the ankle proprioceptive training for patients with DPN, their participants had slower baseline speed (0.71 m/s), smaller cadence (80), similar stride time (1.24 second), smaller step length (0.52m), and smaller double support time (0.32 second) than our baseline data [49]. Another study examines the efficacy of exercise rehabilitation program in improving gait of DPN patients had slower baseline speed (0.71 m/s), smaller cadence (82.94), larger stride time (1.72 second), similar double support time (0.46 second) and ankle range of motion (19.38 degree) than this study [50].”

9.Are there any other clinical parameters to include including history of foot ulcer and HbA1C?

Thank you for the comment/ Patients with a history of foot ulcers were not excluded. In this pilot study we did not collect the HbA1C, but in future studies we will track the HbA1C at both baseline and post-intervention.  We will add this as a limitation of our study which we should address in future studies. Below statements were added to the discussion session:

“In this pilot study, patients with a history of foot ulcers were not excluded, although no participants had active foot ulcers. We did not collect clinical parameters of diabetes severity such as HbA1C. These clinical parameters could impact the effect of FMV on gait. In future studies we will assess the medical record of participants, and track their HbA1C at both baseline and post-intervention.”

10.Ankle range of motion did not improve with intervention. Is this because the ROM was similar to a control sample? The lack of improvement is discounted in the discussion and it is unclear if this is intended to mean that the ankle ROM improvement is hard to detect. It is likely that is was not compromised to begin with. This ties in with (9) above.

Thank you for the comment. This is a great point to explain the non-significant changes on ankle range of motion. Based on the review by Fernando et al., 2013 of diabetic neuropathy and gait, and the systematic review by Hazari et al., 2016 of kinetics and kinematics of diabetic foot in type 2 diabetes with and without neuropathy, patients with DPN have a reduced motion at the ankle in dorsiflexion and plantar flexion. When compared our baseline ankle ROM to Yavuzer et al., 2006, and Gomes et al., 2011, the ROM was similar to the control group in their study. Therefore, we agreed with you that for our sample, the ankle ROM was not compromised at the baseline which explained why no significant improvements were observed after the intervention.

Reviewer 2 Report

The article aims to investigate whether FMV can improve gait in patients with diabetic peripheral neuropathy. The premise and background support for the paper are strong and there is a need for treatments to improve gait in patients with peripheral neuropathy. There are a few weaknesses to the paper and they are detailed below:

  1. Lack of a placebo group. The authors acknowledge this, but it is a big issue and lowers excitement and importance of this paper. Without a placebo group, most or all of the small improvements in gait could be a placebo effect. This limits any conclusive statements as to the benefits of this technique for patients with diabetic peripheral neuropathy.
  2. Lack of data on severity of peripheral neuropathy. Would be good to know the severity of peripheral neuropathy, by Von Frey filaments at ankle or fingers, electrophysiology measurements, or some other quantifiable characteristic, in the different groups. Could be one reason for lack of a difference between groups since severity of peripheral neuropathy may impact response to this treatment.
  3. Should show all data.  Should have a chart with data divided up by Group1, 2, and 3, not just mention the lack of significance in the results section. This was part of the methods and then ALL of data is shown as compilation of 3 groups. Even if not significant, could have trends that are important for readers. 
  4. Is McKinney et al (2015) in Stud. Health. Technol. Inform a similar tactile stimulus?  This was done in peripheral neuropathy and should be referenced.  

Author Response

1.Lack of a placebo group. The authors acknowledge this, but it is a big issue and lowers excitement and importance of this paper. Without a placebo group, most or all of the small improvements in gait could be a placebo effect. This limits any conclusive statements as to the benefits of this technique for patients with diabetic peripheral neuropathy.

Thank you for the comment. This pilot study was exploratory in nature and will inform future study. We acknowledged that there is a lack of control group and appreciated your insight on this.

  1. Currently there is lack of stand care for gait impairment in DPN, which is a progressive degeneration of the peripheral nerves, particularly in the lower limbs. We have shown promise for this intervention with vibration therapy. Currently the DPN symptom of abnormal gait is getting worse without active care or effective intervention. Unfortunately, none of our participants received any active care or interventions for their DPN associated gait abnormities.
  2. We have reviewed articles with control groups on other interventions for gait rehab, for example, exercise, to see how gait of the control group in other studies changed. In studies by Suzuki et al., 2019 both intervention and control group showed significant improvements on the spatiotemporal parameters. However, the control group in their study underwent 2 weeks of supervised rehabilitative treatment (40 min/day) while the intervention group received extra rhythmic auditory stimulation [48]. Similarly, in study by Saleh and Rehab 2014, they reported significant improvement on gait parameters for both intervention and control group with the control group received traditional physical therapy exercise [49]. In the study that control group received no intervention but regular activities, no significant improvement on any gait parameters were reported with some of the parameters got worsen [50].
  3. We still agree with you that a control group or placebo group should be added in future study. Due to the pilot nature and lack of incentives to keep participants in the study for a control group, we did not include a control group in this study. Because participants were excited to use the wearable FMV device; they may have refused to participate the study if assigned to control group.

Below statements were added to the discussion session:

“We reviewed articles with control groups on other interventions for gait rehab, for example, exercise, to see how gait of the control group in other studies changed. In studies by Suzuki et al., 2019 both intervention and control groups demonstrated significant improvements on spatiotemporal parameters. However, the control group in their study underwent two weeks of supervised rehabilitative treatment (40 min/day) while the intervention group received extra rhythmic auditory stimulation [48]. Similarly, Saleh and Rehab reported significant improvement in gait parameters for both the intervention and control groups. However, their control group received traditional physical therapy exercise [49]. In the study that control group received no intervention but regular activities, no significant improvement on any gait parameters were reported and some of the parameters got worse [50].”

2. Lack of data on severity of peripheral neuropathy. Would be good to know the severity of peripheral neuropathy, by Von Frey filaments at ankle or fingers, electrophysiology measurements, or some other quantifiable characteristic, in the different groups. Could be one reason for lack of a difference between groups since severity of peripheral neuropathy may impact response to this treatment.

Thank you for the comments. We did not collect the data on severity of peripheral neuropathy. We agreed that this could be one reason we did not detect significant changes across the three groups. We did notice that the baseline gait parameters were significantly different across the three groups. We will include assessment of severity of peripheral neuropathy (for example, the Michigan Neuropathy Screening Instrument) in our future studies at both baseline and post-intervention, as well as electrophysiology measures for sensation. Below statements were added to the discussion session:

“Furthermore, we did not collect data on severity of peripheral neuropathy. This could contribute to the non-significant changes across the three groups. We will assess the severity of peripheral neuropathy (for example, the Michigan Neuropathy Screening Instrument) in our future studies at both baseline and post-intervention, as well as electrophysiology measures for sensation.”

3.Should show all data.  Should have a chart with data divided up by Group1, 2, and 3, not just mention the lack of significance in the results section. This was part of the methods and then ALL of data is shown as compilation of 3 groups. Even if not significant, could have trends that are important for readers. 

Thank you for the comment. We have revised Table 2 and 3 to include pre- and post- gait data for each group. Actually, after carefully review the results based on individual groups, we observed that even though not significant, we observed larger changes on the gait parameters in Group 1 than 2 than 3.

We added those statements in the discussion session to explain the differences in the changes:

“After intervention, we observed the changes in the gait parameters were larger in group 1, then group 2 and 3. We hypothesize that this may be because patients with DPN are experiencing sensory loss, particularly vibratory sense. When the continuous vibration was delivered in group 1, during initial use, it might work well but when the participants get used to the vibration, the effect was reduced or diminished. Alternatively, groups 1 and 2, who used to change vibratory pattern frequencies may have taken longer for accommodation to the vibration patterns especially group 1 who used sinusoidal vibration. Future study will be needed to test the hypothesis and if true, then the best vibration delivery could be randomized frequencies within a safe and effective range.”

4.Is McKinney et al (2015) in Stud. Health. Technol. Inform a similar tactile stimulus?  This was done in peripheral neuropathy and should be referenced.  

Thank you for the suggestion. We added this to the background session.

“McKinney et al. assessed the impact of a wearable tactile feedback system for gait rehabilitation in patients with non-diabetic peripheral neuropathy. Even though the stimulus in the study was only used as a feedback system most participants could sense the vibration [32]. While FMV provides benefits to both the peripheral system and neuromotor system,”

Round 2

Reviewer 1 Report

The revised paper has clarified many unclear areas in the manuscript. The addition of comparing the assistive devices is interesting and it is recommended that the discussion reflect the trends observed. For example, AT showed a trend to improve dorsiflexion compared to the non-AT group. The improvement of knee flexion was driven by the changes in knee flexion. This highlights the differences in gait between different types of patients. 

The additional information on physical activity (that several reported walking more) while anecdotal is quite encouraging. It is recommended that the exact number of subjects be reported. 

The paragraph on greater improvements between groups is not entirely founded, consider reviewing and revising.

Minor: With the additional Table on AT vs non-AT, there are now two tables labeled Table 3. Also, an error in spelling for Left "peal" knee flexion. 

Author Response

The revised paper has clarified many unclear areas in the manuscript. The addition of comparing the assistive devices is interesting and it is recommended that the discussion reflect the trends observed. For example, AT showed a trend to improve dorsiflexion compared to the non-AT group. The improvement of knee flexion was driven by the changes in knee flexion. This highlights the differences in gait between different types of patients. 

Thank you for the comments and suggestion. We added below text in the discussion session regarding the changes observed in the AT and non-AT group:

“While both AT and non-AT users showed significant improvements, we observed greater changes in most of the spatiotemporal parameters in AT users than non-AT users. This could because that at the baseline, the AT users were walking slower with more compromised parameters. Given the wearable FMV does not need much active involvement with the users during the treatment, it could be more a viable option for those DPN patients with limited functions to participant other exercise interventions. AT users group showed a trend to improve dorsiflexion compared to the non-AT group. The improvement of knee flexion was driven by the changes in knee flexion. This highlights the differences in gait between different types of patients. In the future study we will improve the current method in gait data collection to be able to analyze gait kinetics for AT users, which will allow us to further learn how FMV works for AT users.”

The additional information on physical activity (that several reported walking more) while anecdotal is quite encouraging. It is recommended that the exact number of subjects be reported. 

Thank you for the suggestion. We added this statement with the number of subjects reported changes of physical activity.

“Eight participants commented they walked more or were more active after two weeks of vibration therapy”

The paragraph on greater improvements between groups is not entirely founded, consider reviewing and revising.

Thank you for the comment. We changed the paragraph in the paper as shown below:

“After intervention, we observed the changes in the gait parameters were larger in group 1 with the sinusoidal vibration and 2 with the pulsing vibration than group 3 with continuous vibration as shown in Figure 2 and 3. This could because the overall better baseline values in most gait parameters in group 3. It could also be possible that DPN patients responded differently to three type of vibration modes. Here we hypothesize the later as patients with DPN are experiencing sensory loss, particularly vibratory sensation. When the continuous vibration was delivered in group 3, during initial use, it might work well but when the participants get used to the vibration, the effect was reduced or diminished. Alternatively, groups 1 and 2 with the changing vibratory frequencies may have taken longer for accommodation to the vibration pattern. Future study will be needed to test the hypothesis and if true, then the best vibration delivery could be randomized frequencies within a safe and effective range.”

Minor: With the additional Table on AT vs non-AT, there are now two tables labeled Table 3. Also, an error in spelling for Left "peal" knee flexion. 

Thank you. We changed the table for the AT vs non-AT group to Table 4.

Reviewer 2 Report

The paper is improved by the text changes. The new paragraph (starting line 273) about differences between the three groups is problematic. Not clear why they conclude 1 is better than 2 and 3 (in my estimation 1 and 2 are better than 3). Please justify this with data, not just a subjective sense. Also, may have a typo in line

"After intervention, we observed the changes in the gait parameters were larger in group 1, then group 2 and 3. We hypothesize that this may be because patients with DPN are experiencing sensory loss, particularly vibratory sense. When the continuous vibration was delivered in group 1, during initial use, it might work well but when the participants get used to the vibration, the effect was reduced or diminished. Alternatively, groups 1 and 2, who used to change vibratory pattern frequencies may have taken longer for accommodation to the vibration patterns especially group 1 who used sinusoidal vibration. 

Do the authors mean "alternatively, groups 2 and 3..."?

Author Response

The paper is improved by the text changes. The new paragraph (starting line 273) about differences between the three groups is problematic. Not clear why they conclude 1 is better than 2 and 3 (in my estimation 1 and 2 are better than 3). Please justify this with data, not just a subjective sense. Also, may have a typo in line "After intervention, we observed the changes in the gait parameters were larger in group 1, then group 2 and 3. We hypothesize that this may be because patients with DPN are experiencing sensory loss, particularly vibratory sense. When the continuous vibration was delivered in group 1, during initial use, it might work well but when the participants get used to the vibration, the effect was reduced or diminished. Alternatively, groups 1 and 2, who used to change vibratory pattern frequencies may have taken longer for accommodation to the vibration patterns especially group 1 who used sinusoidal vibration. “

Do the authors mean "alternatively, groups 2 and 3..."?

Thank you for the comments. We agreed with you that the changes in the gait parameters between group 1 and 2 are negligible (changes in group 1 were slightly higher than group 2) but both showed more improvement than group 3. We revised the paragraph as shown below:

“After intervention, we observed the changes in the gait parameters were larger in group 1 with the sinusoidal vibration and 2 with the pulsing vibration than group 3 with continuous vibration as shown in Figure 2 and 3. This could because the overall better baseline values in most gait parameters in group 3. It could also be possible that DPN patients responded differently to three type of vibration modes. Here we hypothesize the later as patients with DPN are experiencing sensory loss, particularly vibratory sensation. When the continuous vibration was delivered in group 3, during initial use, it might work well but when the participants get used to the vibration, the effect was reduced or diminished. Alternatively, groups 1 and 2 with the changing vibratory frequencies may have taken longer for accommodation to the vibration pattern. Future study will be needed to test the hypothesis and if true, then the best vibration delivery could be randomized frequencies within a safe and effective range.”